# Regulation of nuclear transcription by mitochondrial RNA in endothelial cells

Kiran Sriram[1,2†], Zhijie Qi[3†], Dongqiang Yuan[1], Naseeb Kaur Malhi[1], Xuejing Liu[1], Riccardo Calandrelli[3], Yingjun Luo[1], Alonso Tapia[1,2], Shengyan Jin[4], Ji Shi[5], Martha Salas[6], Runrui Dang[7], Brian Armstrong[6], Saul J Priceman[8], Ping H Wang[9], Jiayu Liao[7], Rama Natarajan[1,2], Sheng Zhong[3*], Zhen Bouman Chen[1,2*]

[1]Department of Diabetes Complications and Metabolism, City of Hope, Duarte, United States; [2]Irell and Manella Graduate School of Biological Sciences, City of Hope, Duarte, United States; [3]Department of Bioengineering, University of California San Diego, La Jolla, United States; [4]Department of Genetics, Yale University School of Medicine, New Haven, United States; [5]Translura, Inc, New Haven, United States; [6]Department of Stem Cell Biology and Regenerative Medicine, City of Hope, Duarte, United States; [7]Department of Bioengineering, University of California Riverside, Riverside, United States; [8]Department of Hematology & Hematopoietic Cell Transplantation, Department of Immuno-oncology, City of Hope, Duarte, United States; [9]Department of Diabetes, Endocrinology, and Metabolism, City of Hope, Duarte, United States

**\*For correspondence:**
szhong@ucsd.edu (SZ);
zhenchen@coh.org (ZBC)

[†]Co-first authors

**Abstract** Chromatin-associated RNAs (caRNAs) form a relatively poorly recognized layer of the epigenome. The caRNAs reported to date are transcribed from the nuclear genome. Here, leveraging a recently developed assay for detection of caRNAs and their genomic association, we report that mitochondrial RNAs (mtRNAs) are attached to the nuclear genome and constitute a subset of caRNA, thus termed mt-caRNA. In four human cell types analyzed, mt-caRNAs preferentially attach to promoter regions. In human endothelial cells (ECs), the level of mt-caRNA–promoter attachment changes in response to environmental stress that mimics diabetes. Suppression of a non-coding mt-caRNA in ECs attenuates stress-induced nascent RNA transcription from the nuclear genome, including that of critical genes regulating cell adhesion, and abolishes stress-induced monocyte adhesion, a hallmark of dysfunctional ECs. Finally, we report increased nuclear localization of multiple mtRNAs in the ECs of human diabetic donors, suggesting many mtRNA translocate to the nucleus in a cell stress and disease-dependent manner. These data nominate mt-caRNAs as messenger molecules responsible for mitochondrial–nuclear communication and connect the immediate product of mitochondrial transcription with the transcriptional regulation of the nuclear genome.

## Editor's evaluation

This work is fundamental in providing compelling evidence of mitochondria-encoded RNAs playing a role in controlling nuclear gene expression. How mitochondria and the nucleus communicate is an important yet, not well-appreciated area of biology. Using the iMARGI (in situ mapping of RNA-Genome Interactions) technology developed by this team, the authors found that mitochondria-encoded RNAs play an unexpected role in regulating nuclear gene expressions in endothelial cells and intriguingly, depletion or overexpression of a specific mt-caRNA altered stress-induced transcription of nuclear genes encoding for innate inflammation and endothelial activation. Overall, these findings are interesting and supported by experimental confirmation, bulk-RNA-seq, and

snRNA and scRNA-seq data and will be of interest to the field studying RNA regulation, gene expression, and cell biology.

## Introduction

Mitochondria are key cellular organelles that control energy production, stress response, and cell fate. Mitochondrial fitness is paramount to cellular homeostasis and organismal health, and its dysfunction is a common denominator of numerous diseases ranging from metabolic, cardiovascular, to neurological disorders (*Wallace, 2005*). As derivatives of endosymbiotic bacterial ancestors, mitochondria in eukaryotes possess their own circular and double-stranded genome (mtDNA) of 16.5 kb in size. mtDNA is bidirectionally transcribed to produce 13 mRNAs, 2 rRNAs, and 22 tRNAs involved in oxidative phosphorylation (OXPHOS). The mitochondrial genome also produces non-coding RNAs (ncRNAs) including small RNAs and long ncRNAs (lncRNAs), whose functions are yet to be fully characterized (*Mercer et al., 2011*; *Rackham et al., 2011*; *Ro et al., 2013*).

As a critical hub for cellular processes and signaling pathways, mitochondria coordinate with the nucleus to regulate gene expression and cell function (*Quirós et al., 2016*). On the one hand, the nucleus communicates to the mitochondria through anterograde signaling. For example, nucleus-encoded transcription factors (TFs) such as NRF1 and TFAM, co-regulators such as PGC1α, and the RNA polymerase POLRMT regulate mitochondrial biogenesis, mtDNA replication, and transcription (*Gleyzer et al., 2005*; *Virbasius and Scarpulla, 1994*; *Ojala et al., 1981*; *Falkenberg et al., 2002*; *Kühl et al., 2016*). Nucleus-transcribed non-coding RNAs (ncRNAs) have also been shown to regulate mitochondrial RNA (mtRNA) processing and translation (*Vendramin et al., 2018*; *Puranam and Attardi, 2001*; *Zhang et al., 2014*). Conversely, mitochondria can communicate with nucleus through mitochondrial retrograde signaling, typically mediated by reactive oxygen species, ATP, and $Ca^{2+}$ (*De Stefani et al., 2016*; *Liu and Butow, 2006*; *Vizioli et al., 2020*). Mitochondria-derived short peptides have also been shown to regulate nuclear gene expression (*Reynolds et al., 2021*). Moreover, mitochondria DNA (mtDNA) and double-stranded RNAs (dsRNAs) can also be released into the cytoplasm, which turn on an endogenous 'danger signal' to trigger a type-I interferon response and innate immune surveillance (*Huang et al., 2020*; *Maekawa et al., 2019*; *West et al., 2015*; *Dhir et al., 2018*; *Tigano et al., 2021*). However, it is unclear whether the immediate output of mitochondrial transcription, that is, the mtRNAs, can act as retrograde signaling messengers to communicate directly with the nucleus.

Several mtRNAs have been detected in the nucleus. Burzio et al. detected *MT-RNR2*-derived sense non-coding mitochondrial RNA (SncmtRNA) in the nucleus of human endothelial cells (ECs) and several cancer cell lines (*Villegas et al., 2007*; *Burzio et al., 2009*; *Landerer et al., 2011*). Perry et al. identified a small mitochondrial-encoded ncRNA, mito-ncR-805, localized in the nucleus of alveolar epithelial cells, and correlated with the increased expression of nuclear-encoded genes that regulate mitochondrial function (*Blumental-Perry et al., 2020*). Yet to be clarified is whether any nuclear-translocated mtRNA can alter the epigenome, and subsequently impact gene transcription and cellular behavior.

The chromatin-associated RNAs (caRNAs) form a critical layer of the epigenome that can regulate nuclear transcription (*Li and Fu, 2019*; *Nguyen et al., 2018*). Here, we show that caRNAs are not exclusively nuclear genome-produced RNA – they also include mtRNA. The chromatin-associated mtRNAs (mt-caRNAs) preferentially attach to the promoter regions of the nuclear genome. In human ECs, the mtRNA–chromatin attachment levels change in response to cellular stress induced by high glucose and tumor necrosis factor alpha (TNFα) (HT). As an example, we identify the mitochondrial lncRNA SncmtRNA as a mt-caRNA. In human ECs, suppression of SncmtRNA attenuates HT stress induction of nascent RNAs transcribed from the nuclear genome, including the cell adhesion molecules ICAM1 and VCAM1, and abolishes stress-induced monocyte–EC adhesion. In addition to SncmtRNA, we show nuclear localization of other mtRNAs including MT-CYB and MT-ND5 in human ECs, which is increased by HT and in diabetic human donors. Collectively, our findings suggest that mtRNAs are a class of retrograde messengers that mediates cellular response by attaching to chromatin and regulating nuclear transcription.

## Results

### Pervasive association of mtRNA with chromatin

To test whether mtRNAs are attached to the nuclear genome, we generated high-resolution RNA–genome interaction maps from human embryonic stem cells (H1), foreskin fibroblasts (HFF), and lymphoblasts (K562) (*Calandrelli et al., 2021*) using our recently developed iMARGI (in situ mapping of RNA–genome interactions) technology (*Yan et al., 2019*; *Sridhar et al., 2017*). By simultaneously sequencing caRNAs and their associated genomic DNA sequences from purified nuclei (*Wu et al., 2019*), iMARGI can differentiate the sequencing reads originating from RNA (RNA-end reads) and genomic DNA (DNA-end reads) in the chimeric reads. As expected, iMARGI data revealed diverse caRNAs transcribed from thousands of coding and non-coding genes as well as transposable elements in the nuclear genome (*Calandrelli et al., 2021*).

Surprisingly, in each cell type profiled, more than 6% of iMARGI-mapped read pairs had the RNA-end reads uniquely mapped to mtRNAs. These large numbers of mtRNA reads cannot be due to errors mapping to the nuclear DNA of mitochondrial origin (NUMT) (*Lopez et al., 1994*), because 98.7% of mtRNA-matching sequences contain more than 5 bp mismatches to any NUMTs. Based on the normalized RNA-end reads, mtRNAs exhibited higher chromatin association level (CAL) than that of nuclear genome-transcribed RNAs (nuRNAs) in all three cell types (p = 0.0022, *t*-test) (*Figure 1A*). iMARGI also revealed the attachment of mtRNAs to various genomic regions, including promoters, coding sequences (CDS), 5′ and 3′ untranslated regions (UTRs), enhancers, super enhancers (SEs), introns, and intergenic regions, with promoters showing the highest enrichment of mtRNA association in all three cell types (largest p = 0.0074, one-way analysis of variance [ANOVA]) (*Figure 1B–D*). SncmtRNA (GenBank: DQ386868.1), a mitochondria-encoded lncRNA that has been observed to be nuclear localized (*Villegas et al., 2007*; *Burzio et al., 2009*), is also one of the caRNAs and showed higher CALs than that of nuRNA (p-value = 0.00017, Student's *t*-test) (*Figure 1A*) and enriched association with promoters as compared to other genome regions (*Figure 1E–G*).

Next, we investigated whether the chromatin–mtRNA association changes under a cellular stress condition. To this end, we analyzed iMARGI data from human umbilical vein ECs (HUVECs) under normal glucose (NM) and combined treatment of high glucose and TNFα (HT), which we have used to mimic healthy vs diabetic conditions (*Calandrelli et al., 2020*). While ECs under NM control showed comparable levels of chromatin-attached mtRNAs as the other three cell types, the CALs of mtRNA increased from NM to HT (p-value = 0.037, Student's *t*-test) (*Figure 1H*). In contrast, under both conditions, read pairs between nuRNA and mitochondrial DNA (mtDNA) or mtRNA and mtDNA were barely detected (*Figure 1H*), suggesting minimal mitochondrial contamination in the iMARGI libraries.

Quantification of the CAL of mtRNAs, including coding and non-coding transcripts, indicated that the majority of mtRNAs exhibit not only abundant but also higher levels of chromatin association than the nuRNAs under both conditions in ECs (*Figure 1I, J*). Moreover, comparison between NM and HT revealed that HT further induced the CALs of most of the mtRNAs (*Figure 1K*). Specifically, SncmtRNA exhibited a 64-fold higher CAL than the average CAL of all nuRNAs in NM (p = 0.00082, Student's *t*-test) (*Figure 1I*), which was further increased by HT (p = 0.034, Student's *t*-test) (*Figure 1K*). Taken together, these data identify mtRNA as a component of caRNA (hereafter called mt-caRNA) with preferential genomic association with promoters in multiple cell types. Furthermore, the mtRNA–chromatin association can be affected by a stress condition, exemplified by ECs under HT and numerous mtRNAs including SncmtRNA.

Given that HT induces significant gene expression changes in ECs (*Calandrelli et al., 2020*), we asked if mt-caRNAs is enriched at the promoters of genes that respond to HT stress conditions. To this end, we identified 312 HT-induced genes in ECs from RNA-seq (adjusted p-value <0.05). Compared with other gene promoters, and other genomic regions including enhancers, SEs, introns, 5′ and 3′ UTRs, CDS, and intergenic regions, the promoters of HT-induced genes exhibited higher CAL of mtRNAs (p = 0.029, Student's *t*-test) (*Figure 1L, M*) and SncmtRNA (p = 0.016, Student's *t*-test) (*Figure 1N, O*). We observed the same behaviors for both mtRNA and SncmtRNA under HT condition with slightly higher statistical significance (p = 0.025 for mtRNA, p = 0.0087 for SncmtRNA, Student's *t*-test) (*Figure 1L, O*). These results indicate that mt-caRNAs including SncmtRNA are indeed enriched at the promoters of HT-induced genes under both normal and HT conditions.

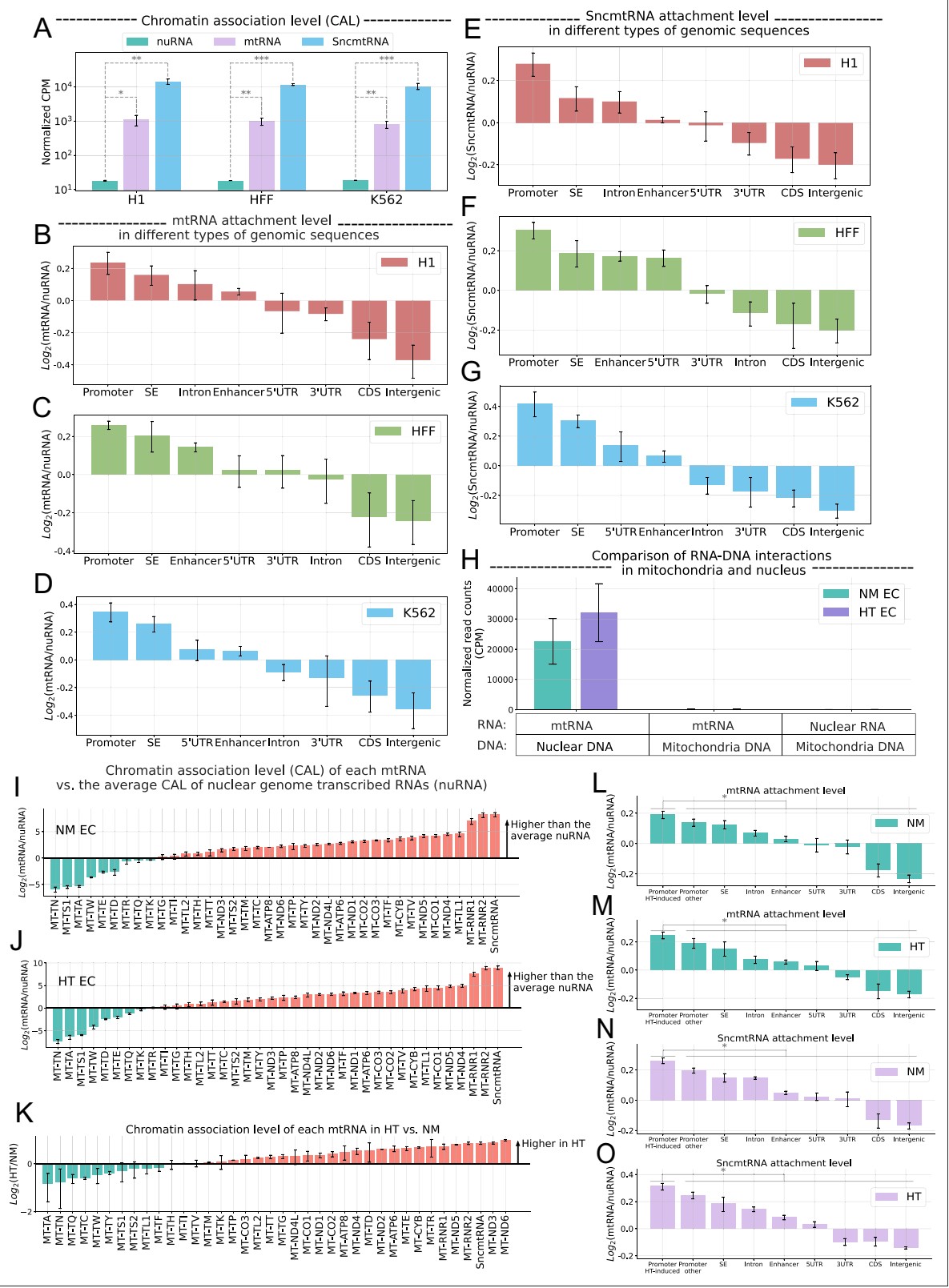

**Figure 1.** Effect of SncmtRNA KDAssociation of mtRNA with chromatin. (**A**) Chromatin association levels (CAL) of nuclear genome-transcribed RNA (nuRNA) (blue), mtRNA (green), and the SncmtRNA (purple) in H1 embryonic stem cells (H1), foreskin fibroblasts (HFF), and K562 lymphoblasts (K562) (columns). CAL is estimated by the normalized counts per million (CPM) of iMARGI RNA-end read counts. *p-value<0.05, **p-value <0.01, ***p-value <0.001. Normalized CAL of mtRNA (*y*-axis) on eight types of genomic sequences (columns) in H1 (**B**), HFF (**C**), and K562 (**D**) Log₂(mtRNA/nuRNA): the

*Figure 1 continued on next page*

*Figure 1 continued*

log ratio of mtRNA and nuRNA read counts in the iMARGI RNA-end reads. SE: super enhancer; UTR: untranslated region; CDS: coding sequence. Normalized CAL of the SncmtRNA (*y*-axis) on eight types of genomic sequences (columns) in H1 (**E**), HFF (**F**), and K562 (**G**) Log$_2$(SncmtRNA/nuRNA): the log ratio of SncmtRNA and nuRNA read counts in the iMARGI RNA-end reads. (**H**) Normalized numbers of iMARGI read pairs (*y*-axis) mapped to mtRNA and nuclear DNA (first two columns), mtRNA and mitochondrial DNA (middle two columns), and nuRNA and mitochondrial DNA (last two columns) in normal mannitol control (NM)- (green) and high glucose and TNFa (HT)-treated (purple) endothelial cells (ECs). Comparison of the CAL of each mtRNA (column) and the average CAL of all nuRNA in NM (**I**) and HT EC (**J**). *y*-Axis: the log ratio of mtRNA and nuRNA read counts in the iMARGI RNA-end reads. $y > 0$ indicates that this mtRNA (column) exhibits a higher level of chromatin association than the average CAL of the nuRNA. (**K**) Comparison of the CAL of each mtRNA (column) between NM- and HT-treated EC. Log$_2$(HT/NM): log ratio of CAL between HT and NM per each mtRNA. $y > 0$ indicates higher CAL in HT vs NM. Normalized CAL of mtRNAs (**L, M**) and SncmtRNA (**N, O**) on different types of genomic sequences in ECs under NM or HT. Promoter HT-induced: the promoters of 312 HT-induced upregulated genes. Log$_2$(mtRNA/nuRNA): the log ratio of mtRNA and nuRNA read counts in the iMARGI RNA-end reads. Data plotted from *n* = 2 biological replicates.

## Effect of SncmtRNA perturbation on the expression and function of nuclear-encoded genes in ECs

The chromatin association of mtRNAs, especially with promoters, prompted us to posit the involvement of mt-caRNAs in nuclear transcription. To test this hypothesis, we elected SncmtRNA in ECs as an example, considering that (1) SncmtRNA show significantly enriched CAL in ECs, which is further increased by HT in the iMARGI data (*Figure 1I–K*) and (2) perturbation of a lncRNA is less likely to directly impact mitochondrial function than targeting a mitochondria-encoded mRNA, rRNA, or tRNA.

We first confirmed the presence and chromatin association of SncmtRNA in ECs. As described (*Villegas et al., 2007*; *Burzio et al., 2009*), SncmtRNA is comprised of 1559 nt sequence encoding mitochondrial 16S rRNA (transcribed from the H strand of mitochondrial genome), with its 5' end appended to an 815 nt invert repeat, forming a putative double-stranded hairpin-shaped structure (*Figure 2A*, *Figure 2—figure supplement 1A*). Based on this description, we verified the presence of SncmtRNA in ECs using reverse transcription (RT)-polymerase chain reaction (PCR) with primers flanking the putative hairpin loop and the unique junction region (*Figure 2A, B*), followed by Sanger sequencing (*Figure 2—figure supplement 1B*). As a control, PCR without RT failed to amplify SncmtRNA, evident of an RNA. Moreover, in Rho$^0$ cells that are depleted of mtDNA (*Chandel and Schumacker, 1999*), SncmtRNA, like 16S rRNA, was not detected (*Figure 2B*), indicative of its mitochondrial origin.

We also performed fluorescence in situ hybridization (FISH) with an antisense locked nucleic acid (LNA) probe targeting the unique junction region of SncmtRNA, which detected abundant signals in the nucleus in ECs. FISH also detected SncmtRNA signals in the cytoplasm resembling the morphology of mitochondria (*Figure 2C*). As an FISH control, probe with scrambled sequence did not reveal specific signals (*Figure 2—figure supplement 1C*). Consistently, qPCR of SncmtRNA in different EC subcellular fractions detected the majority of SncmtRNA in the nuclear fractions including chromatin and nucleoplasm, with MALAT1 (predominantly nuclear localized), GAPDH (predominantly cytoplasm localized), and TUG1 (localized in both nucleus and cytoplasm) quantified as controls (*Figure 2D*). Moreover, compared to NM, HT-increased chromatin-associated SncmtRNA in ECs, while there was no change for MALAT1, TUG1, and GAPDH (*Figure 2D*).

The HT-increased mt-caRNA, including SncmtRNA could be due to the increased total levels of mtRNAs in ECs. Arguing against this possibility, high glucose and TNFα are known to impair mitochondrial biogenesis (*Shenouda et al., 2011*; *Trudeau et al., 2010*). Consistently, bulk RNA-seq of ECs under NM vs HT revealed that several nuclear-encoded genes regulating mitochondrial biogenesis, such as peroxisome proliferator-activated receptor gamma (PPARG) and Transcription Factor A, Mitochondrial (*TFAM*) were downregulated (*Figure 2—figure supplement 1D*). Additionally, mitochondria visualized by Mitotracker showed fragmentation under HT (*Figure 2—figure supplement 1E*). In terms of mitochondria-encoded transcripts, the majority remained unchanged (*Figure 2—figure supplement 1F*). Similarly, the total RNA levels of SncmtRNA as well as 16S rRNA and antisense mitochondrial non-coding RNA (ASncmtRNA), all derived from *MT-RNR2*, were comparable between NM and HT. In contrast, the expression of endothelial nitric oxide synthase (eNOS), hallmark of EC homeostasis was decreased, and that of VCAM1 and ICAM1, typical inflammation markers were strongly induced by HT (*Figure 2E*). Therefore, the pervasive increase of mt-caRNAs is unlikely due to a global increase of total mtRNAs.

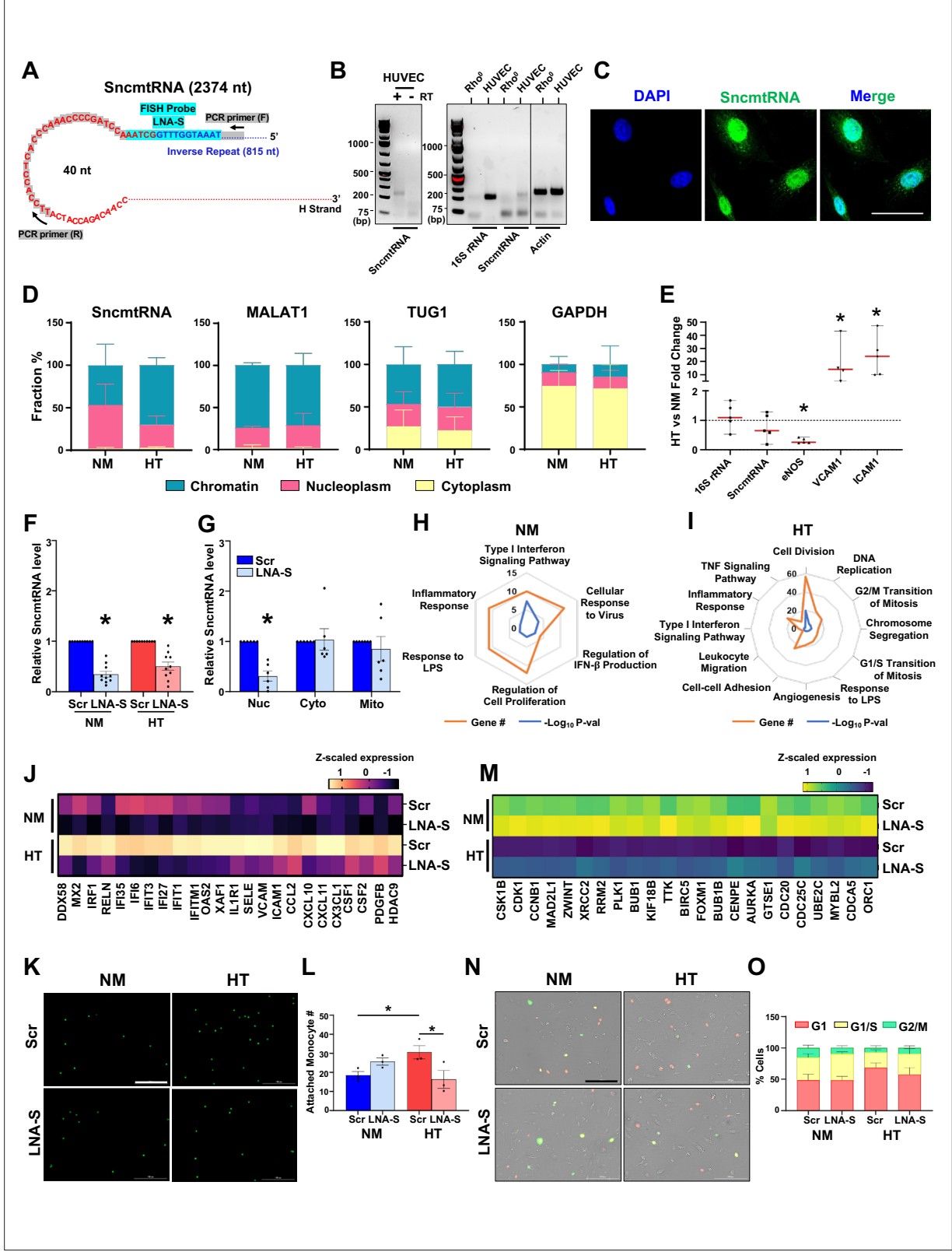

**Figure 2.** SncmtRNA knockdown affects endothelial cell (EC) expression and function. (**A**) Putative SncmtRNA structure and the location of LNA-GapmeR target region (LNA-S) and PCR primers. (**B**) PCR of SncmtRNA. Left: ECs with (+) or without (−) reverse transcription (RT). Right: ECs in comparison with Rho⁰ cells, with 16S rRNA and β-actin detected as controls. (**C**) Fluorescence in situ hybridization (FISH) images of SncmtRNA in ECs, with 4',6-diamidino-2-phenylindole (DAPI) staining nuclei. Scale bar = 50 μm. (**D, E**) ECs were maintained in normal glucose + 25 mM mannitol (NM)

*Figure 2 continued on next page*

*Figure 2 continued*

or high glucose + TNFα (HT) for 72 hr. (**D**) RT-qPCR of respective RNAs in subcellular fractions. Data plotted as mean ± standard error of the mean (SEM) from three independent experiments. (**E**) RT-qPCR of various RNAs in whole cells, with levels in NM set at 1. Data represent mean ± SEM from five independent experiments. RT-qPCR of SncmtRNA in (**F**) total ECs transfected with scramble (Scr) or SncmtRNA-targeting (LNA-S) LNA GapmeRs under NM and HT and in (**G**) subcellular fractions (nucleus, cytoplasm, and mitochondria) from ECs transfected with Scr or LNA-S and kept under NM. Data plotted as mean ± SEM from six independent experiments. (**H–O**) ECs treated as in (**F**) were profiled by bulk RNA-seq (n = 2 biological replicates). (**H, I**) Radar plots showing Gene Ontology (GO) terms enriched in differentially expressed genes (DEGs) due to Snc-KD under NM or HT. Each vertex indicates the enriched GO term. Orange vertex indicates the number of DEGs enriched in the given GO terms and blue vertex indicates $-\log_{10}$ p value. Heatmap showing z-scaled expression in transcripts per million (TPM) of innate immune and inflammatory genes induced by HT and suppressed by Snc-KD in (**J**) and cell cycle genes suppressed by HT and rescued upon Snc-KD in (**M**). (**K, L, N, O**) Representative images and quantification of fluorescently labeled monocytes adhesion to human umbilical vein endothelial cells (HUVECs) (in K, L) and fluorescence ubiquitination cell cycle indicator (FUCCI) assay showing ECs in various phases of cell cycle (in **N, O**). Scale bar = 200 μm. Data plotted as mean ± SEM from three and five independent experiments. *p < 0.05 as compared to NM (in E) or Scr (in **F, G**) or as indicated (in **L**).

The online version of this article includes the following source data and figure supplement(s) for figure 2:

**Source data 1.** Uncropped gel image for *Figure 2B* (left panel): SncmtRNA PCR product in the presence (+) or absence (−) of reverse transcriptase (RT).

**Source data 2.** Uncropped gel image for *Figure 2B* (right panel): reverse transcription (RT)-PCR products of 16S rRNA, SncmtRNA, and β-actin generated using total RNA isolated from Rho$^0$ cells or human umbilical vein endothelial cells (HUVECs).

**Source data 3.** Raw gel images for *Figure 2B*.

**Figure supplement 1.** MT-RNR2-derived SncmtRNA and HT-regulated mitochondrial morphology and transcription.

**Figure supplement 2.** Effect of SncmtRNA KD.

**Figure supplement 3.** Effect of SncmtRNA KD on mitochondrial function.

**Figure supplement 4.** HT-enriched SncmtRNA chromatin attachment likely contributes to transcriptional induction of HT-induced genes.

**Figure supplement 5.** Effect of SncmtRNA-KD in lipopolysaccharide (LPS)-induced genes.

Upon confirming the presence, sequence, and subcellular localization of SncmtRNA, we designed three SncmtRNA-targeting LNA GapmeRs, which can cause RNase H-mediated RNA degradation in the nucleus (*Liang et al., 2017*). While LNA-S1 targets the unique chimeric junction in SncmtRNA, LNA-S2 and S3 target different regions of H strand portion (*Figure 2—figure supplement 2A*). As expected, only LNA-S1 (hereafter termed LNA-S) consistently decreased SncmtRNA level without affecting those of 16S rRNA and ASncmtRNAs (*Figure 2—figure supplement 2B*) and was thus used in the subsequent SncmtRNA knockdown (Snc-KD) experiments. While LNA-S decreased the total cellular SncmtRNA in ECs under both NM and HT (*Figure 2F*), the inhibitory effect was primarily in the nucleus (*Figure 2G*). This is likely because that LNA GapmeRs are predominantly localized in the nucleus and cytoplasm (*Bennett et al., 1992*; *Castanotto et al., 2015*). Their localization in mitochondria is unlikely in the absence of a carrier such as MITO-porter (*Furukawa et al., 2015*). On the other hand, RNase-H1 is enriched in the nuclear and mitochondrial compartments (*Shen et al., 2014*; *Wu et al., 2013*). Thus, the transfection of LNA-S using lipofectamine would result in relatively selective effect in the nucleus. Therefore, any consequence observed with LNA-S should be primarily due to the reduction of SncmtRNA in the nucleus, including chromatin-associated SncmtRNA.

Bulk RNA-seq revealed that under NM, Snc-KD resulted in 265 differentially expressed genes (DEGs), including 160 down- and 105-upregulated (*Figure 2—figure supplement 2C*). Under HT, Snc-KD led to more DEGs, that is 732 DEGs (403 down- and 329 upregulated) in ECs (*Figure 2—figure supplement 2D*), with 120 common DEGs between NM and HT (*Figure 2—figure supplement 2E*). Pathway enrichment analysis indicated that innate immune and inflammatory responses are the most consistently affected pathways under both NM and HT, with more pathways involved in innate immune response under NM (e.g. Type I interferon signaling and cellular response to virus and LPS) and more pathways involved in inflammatory response under HT (e.g., TNF signaling, leukocyte migration, and cell–cell adhesion) (*Figure 2H, I*). Moreover, these pathways are mainly enriched in the downregulated DEGs by Snc-KD (*Figure 2—figure supplement 2G, I*). For example, hallmark genes for innate immune and inflammatory activation, including *DDX58*, *MX2*, *VCAM1*, and *ICAM1*, were strongly suppressed by Snc-KD under both NM and HT (*Figure 2J*).

Consistent with the gene expression changes, the HT-induced pro-inflammatory response of ECs assayed by monocyte adhesion, was abrogated by Snc-KD (*Figure 2K and L*). In contrast, cell proliferation, cell division, and cell cycle regulation were mostly upregulated by Snc-KD, especially under

HT (*Figure 2I*, *Figure 2—figure supplement 2H*), evident by the significant induction of cell cycle regulators, for example, *CDK1* and *CCNB1* (*Figure 2M*). Fluorescence ubiquitination cell cycle indicator (FUCCI) assay, which detects cell cycle progression and division (*Sakaue-Sawano et al., 2008*), revealed while HT caused a cell cycle arrest at G1 phase, LNA-S led to a trend toward promoting the G1-arrested cells to S phase (*Figure 2N, O*). In contrast, Snc-KD had no clear effect on mitochondria network or morphology (*Figure 2—figure supplement 3A–F*). Furthermore, Snc-KD led to minimal changes in mitochondrial function as quantified by oxygen consumption rate (OCR) using Seahorse assay (*Figure 2—figure supplement 3G*). Additionally, we did not observe any significant effect on the levels of mtRNAs or those of nuclear-encoded genes regulating mitochondrial biogenesis upon Snc-KD (*Figure 2—figure supplement 3H–K*). Taken together, these data suggest that the mitochondria-encoded and partially chromatin-associated SncmtRNA plays a role in the regulation of nuclear-encoded transcripts.

The enriched CAL of SncmtRNA in the promoter regions of HT-induced genes and the effect of Snc-KD in the nuclear-encoded transcriptome raised the possibility that HT-induced gene transcription is in part through enriched association of SncmtRNA with promoters. Based on the iMARGI and RNA-seq data from ECs under NM vs HT, we identified 14 HT-induced genes that exhibit significant increase in the promoter-associated SncmtRNA (Benjamini–Hochberg adjusted p-value <0.05) (*Figure 2—figure supplement 4A*). To evaluate whether these genes are more likely to be downregulated by Snc-KD, we performed an association (Chi-square) test between these genes and the Snc-KD-reduced genes. Out of these 14 genes, Snc-KD reduced the expression levels of 12 of them under HT (Snc-KD vs Scr fold-change <1), including 6 (*ICAM1*, *ZFHX2*, *ABCA6*, *CSF1*, *PDE5A*, and *RND1*) with statistical significance (Benjamini–Hochberg adjusted p-value <0.05), suggesting the genes suppressed by Snc-KD are enriched in these genes (odds ratio = 2.82, p-value = 0.0096, Chi-square test) (*Figure 2—figure supplement 4B*). This enriched association between genes that show increased SncmtRNA–promoter association and genes that are positively regulated by SncmtRNA supports our hypothesis that HT-induced transcription is likely through enriched SncmtRNA–promoter association.

## SncmtRNA regulates transcription of the nuclear genome

The above described findings further prompted us to test the hypothesis that SncmtRNA regulates the transcription of nuclear-encoded genes. We reasoned that the effect of Snc-KD should be reflected by changes in the levels of nascent RNA, the immediate products of transcription. Furthermore, SncmtRNA transcriptional targets should show consistent changes between the nascent RNA levels in the nuclei and the total RNA levels in whole cells by Snc-KD. Thus, we performed single-nuclear and single-cell RNA-seq (snRNA and scRNA-seq) in ECs with or without Snc-KD and subjected to HT to capture simultaneously the nuclear and the whole-cell transcriptome at single-cell level. While scRNA-seq generated the majority of reads mapped to exons, snRNA-seq generated 40–50% of reads mapped to introns (*Figure 3—figure supplement 1A, B*), enabling better interrogation of nascent RNAs. On average, 5500 cells/sample were sequenced for snRNA-seq and 3500 cells/sample were sequenced for scRNA-seq, both of which performed with biological replicates. While snRNA- and scRNA-seq were highly correlated (Spearman correlation coefficient [SCC] = 0.9) (*Figure 3A–C* and *Figure 3—figure supplement 1C, D*), snRNA-seq identified 462 DEGs (comprising 302 down- and 160 upregulated) and scRNA-seq returned 318 DEGs (comprising 174 down- and 144 upregulated) as a result of Snc-KD in HT (*Figure 3—figure supplement 1E, F*). Considering the comparable average reads per cell and total genes detected by sc- and snRNA-seq, but substantially lower median unique molecular identifier (UMI) counts per cell detected by snRNA-seq (*Figure 3—figure supplement 1G, H*), the nuclear transcriptome appears to show a greater change than the whole-cell transcriptome due to Snc-KO, supporting a role of SncmtRNA in nuclear transcription.

Next, we identified the potential transcriptional targets of SncmtRNA by analyzing DEGs consistent between snRNA- and scRNA-seq datasets. Among common DEGs shared by sc- and snRNA-seq, 100 were downregulated and 33 were upregulated by Snc-KD, suggesting that SncmtRNA is more likely to be a transcriptional activator than suppressor (*Figure 3F*). Among these common DEGs, 43 were consistently downregulated in bulk RNA-seq and sn- and scRNA-seq, including *ICAM1*, *VCAM1*, *CCL2*, and *DDX58* (*Figure 3G*, *Figure 3—figure supplement 2A*). The 43 consistently downregulated genes showed higher correlation between snRNA and scRNA-seq (SCC = 0.65) as compared to other

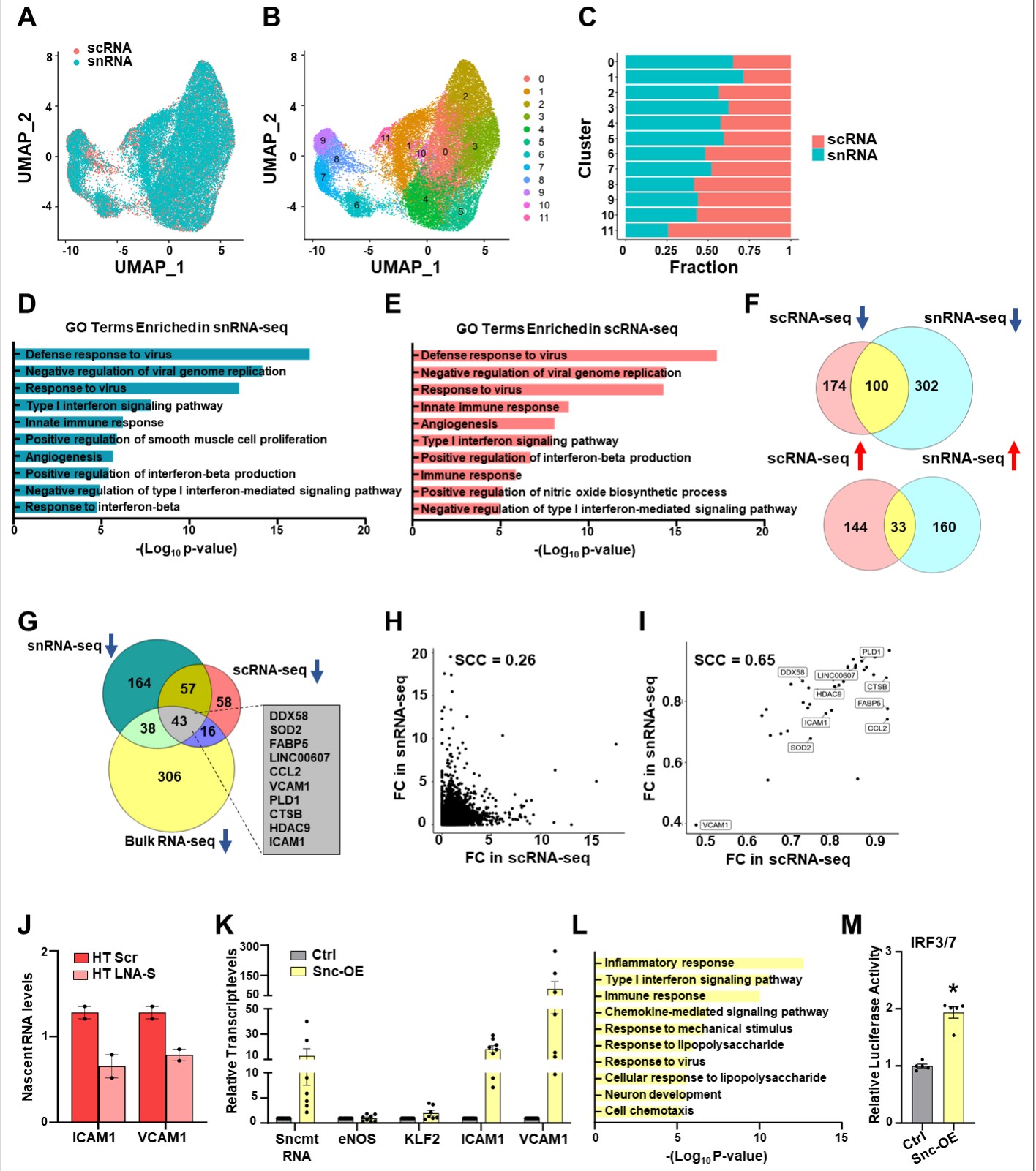

**Figure 3.** SncmtRNA regulates the transcription of nuclear genes. (**A–J**) Endothelial cells (ECs) were transfected with Scr or LNA-S and exposed to HT for 72 hr in two biological replicates and then subjected to scRNA and snRNA-seq (*n* = 2). UMAP embedding colored by cells (scRNA) or nuclei (snRNA) in (**A**) or by unsupervised clustering (**B**). (**C**) Fraction of cells (scRNA) or nuclei (snRNA) in each cluster from (**B**). GO terms enriched by Snc-KD in snRNA-seq (**D**) or scRNA-seq (**E**). (**F**) Venn diagrams showing the number of common down- or upregulated differentially expressed genes (DEGs) between

*Figure 3 continued on next page*

*Figure 3 continued*

sc- and snRNA-seq. (**G**) Venn diagram showing the number of DEGs commonly downregulated by Snc-KD in three RNA-seq datasets. Top ranked DEGs involved in innate immune and inflammatory response among the 43 intersecting genes are shown. Correlation plot between Snc-KD-caused fold change (FC) in sn- and scRNA-seq of the commonly detected genes excluding the 43 intersecting genes (**H**) or the 43 intersecting genes (**I**). SCC are indicated. (**J**) Nascent RNA levels of ICAM1 and VCAM1 were quantified by RT-qPCR in the same batches of ECs used for sc- and snRNA-seq. ECs were infected with control AAV (Ctrl) or AAV-driven SncmtRNA overexpression (Snc-OE) for 48 hr. (**K**) RT-qPCR of respective transcripts (n = 8 biological replicates). (**L**) Enriched pathways (GO terms) by Snc-OE identified from bulk RNA-seq (n = 3 biological replicates). (**M**) Quantification of the activity of luciferase reporter driven by promoter containing IRF3/7 binding sites upon 293T cells infected by AAV (Ctrl) or Snc-OE. *p < 0.05 compared to Ctrl.

The online version of this article includes the following figure supplement(s) for figure 3:

**Figure supplement 1.** Read mapping, correlation, and number of differentially expressed genes (DEGs) of snRNA- and scRNA-seq.

**Figure supplement 2.** Differentially expressed gene (DEG) analysis of snRNA- and scRNA-seq data.

DEGs (SCC = 0.26) (*Figure 3H, I*), suggesting that these genes are more likely to be regulated at the transcriptional level by SncmtRNA. Interestingly, many of these genes are involved in innate immune and inflammatory response, which were the most enriched pathways shared by DEGs identified from snRNA- and scRNA-seq (*Figure 3D, E*). In contrast, 29 genes were consistently upregulated upon Snc-KD in the bulk RNA- and scRNA-seq but not in the snRNA-seq data (*Figure 3—figure supplement 2B, C*). These genes may also be regulated by SncmtRNA but not via transcriptional regulation. Indeed, KLF2 is one such gene and its expression appears primarily regulated post-transcriptionally through miRNA-92a, which occurs in the cytoplasm (*Wu et al., 2011*; *Fang and Davies, 2012*).

To test our hypothesis, we performed nascent RNA pull-down assay to demonstrate that Snc-KD indeed led to a reduction of nascent transcripts of ICAM1 and VCAM1 in ECs exposed to HT (*Figure 3J*). Complementarily, we overexpressed SncmtRNA (Snc-OE) in ECs using AAV9, which enters the nucleus and drives the transcription of SncmtRNA (*Wang et al., 2019*). Snc-OE strongly induced ICAM1 and VCAM1, but not that of eNOS and KLF2 (*Figure 3K*). At a transcriptome level, Snc-OE resulted in 553 DEGs (289 down- and 264 upregulated), among which similar pathways to those affected by Snc-KD, for example Type I interferon signaling and responses to virus and lipopolysaccharide were also enriched (*Figure 3L*). Consistently, Snc-OE in HEK-293 cells also increased the luciferase activity driven by promoters containing binding elements of IRF3/7, key TF mediating Type-I IFN signaling (*Figure 3M*). Collectively, these data suggest that SncmtRNA regulates the transcription of nuclear-encoded genes, exemplified by those promoting innate immune and inflammatory response.

## Nuclear localization of mitochondrial transcripts

To validate our findings on mt-caRNA beyond SncmtRNA, we extended our study to other mt-caRNAs identified from iMARGI analysis. Specifically, we performed single-molecule RNA FISH (smFISH) in ECs for MT-CYB and MT-ND5, which showed high CAL (*Figure 1I–K*) that was further increased by HT in ECs and have sequences feasible for smFISH probe design. We first used RNAscope, which is based on target-specific double Z probes (*Wang et al., 2012*) designed against MT-CYB and MT-ND5, respectively. We ensured that the sequences used for design of RNAscope probes did not match any nuclear genome at 100%, minimizing the possibility of detecting signals from NUMTs. While the positive control (*POLR2A*, a universally expressed gene) detected strong punctate signals and the negative control (*dapB*, a bacterial gene not expressed in humans) yielded no specific signals (*Figure 4—figure supplement 1A, B*), smFISH for MT-CYB and MT-ND5 detected signals largely overlapping with that of Mitotracker (*Figure 4—figure supplement 1C, D*). In contrast, smFISH with Rho$^0$ cells showed no signals of either of the mtRNAs, indicating that the probes indeed detected mitochondrial transcripts (*Figure 4—figure supplement 1E*).

At higher magnification, confocal microscopy revealed a fraction of MT-CYB and MT-ND5 signals in the nuclei as clear puncta, which overlapped with that of DAPI in ECs under NM and significantly increased by HT (*Figure 4A–C*). Three-dimensional reconstruction of the confocal images visualized the MT-CYB signals embedded in the DAPI-stained nuclei in ECs under NM, which became more pervasive by HT (*Figure 4D*). Consistently, super resolution imaging also captured a portion of MT-CYB signals co-stained with DAPI as nuclear puncta in NM-treated ECs, which was remarkably increased after HT treatment (*Figure 4E*).

As an independent validation, we designed separate pools of probes and performed sequential smFISH (*Raj et al., 2008*; *Liu et al., 2020*), which yielded consistent data showing the nuclear

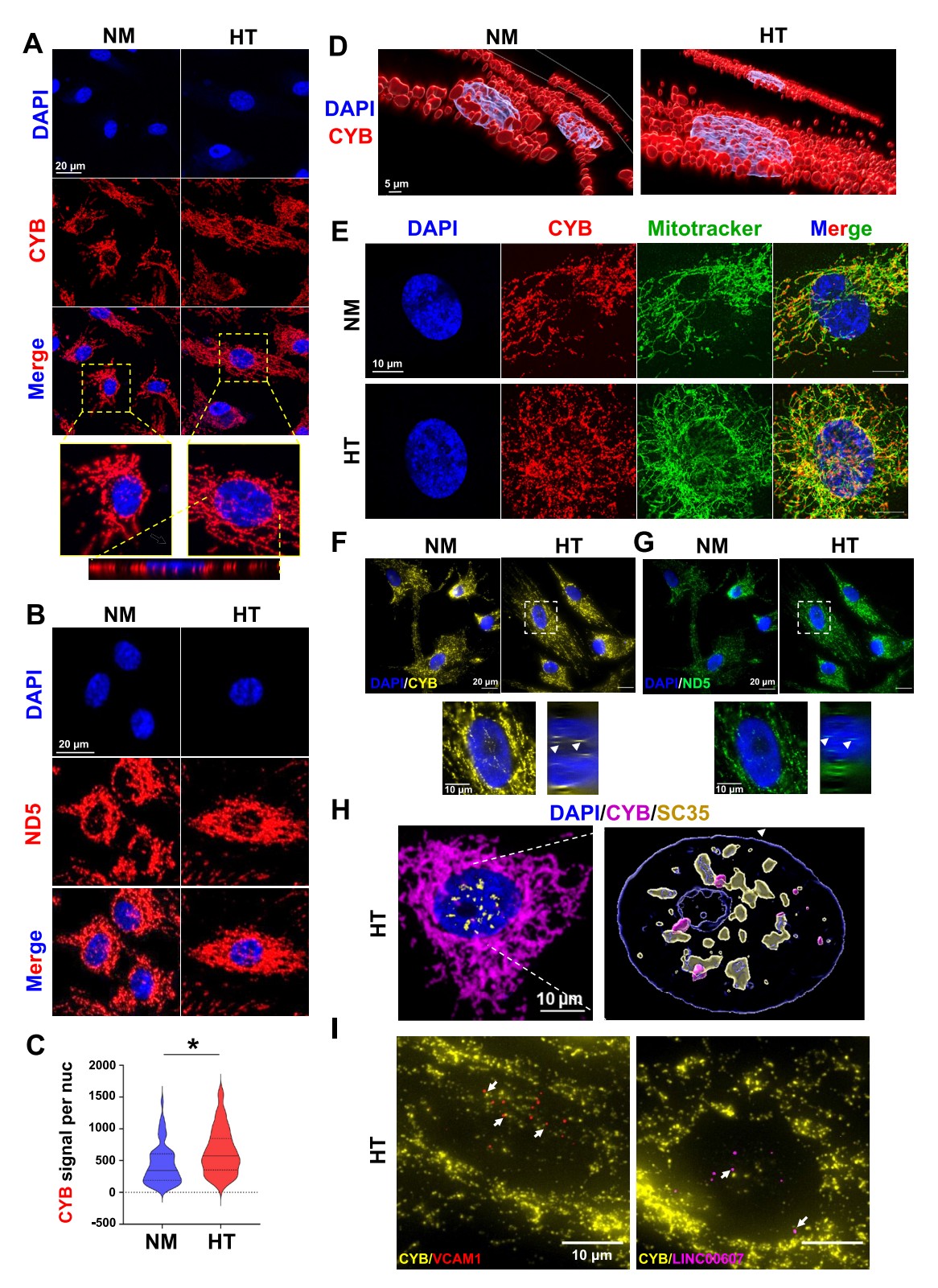

**Figure 4.** Nuclear localization of MT-CYB and MT-ND5. Human umbilical vein endothelial cells (HUVECs) were treated with NM and HT for 72 hr. Confocal images of MT-CYB (**A**) and MT-ND5 (**B**) smFISH in endothelial cells (ECs). Bottom of (**A**): orthogonal view across the vertical axis of HT-EC. (**C**) Violin plot showing the quantification of average MT-CYB signal per nucleus in NM and HT. Over 200 cells from each condition were quantified. *p < 0.05 by Mann–Whitney *U*-test. (**D**) 3D reconstruction of confocal images from (**A**). For (**A–D**) representative images from three independent experiments

*Figure 4 continued on next page*

*Figure 4 continued*

are shown. (**E**) Super resolution microscopic images of MT-CYB smFISH and Mitotracker deep red from a single optical plane of 0.16 μm thickness. (**F, G**) Single-molecule sequential fluorescence in situ hybridization (FISH) with probes for CYB and ND5, with DAPI staining nuclei. (**H**) Super resolution microscopic image of merged signals of smFISH for MT-CYB RNA (in magenta), immunofluorescence for SC35 protein (in yellow), and DAPI (in blue) from a single optical plane of 0.16 μm thickness (left) and the 3D reconstruction (right). (**I**) Co-sm-seqFISH for indicated pre-mRNA transcripts with MT-CYB transcript. (**H**) and (**I**) show images taken from HT-treated ECs.

The online version of this article includes the following figure supplement(s) for figure 4:

**Figure supplement 1.** Control experiments of smFISH for MT-CYB and MT-ND5.

localization of MT-CYB and MT-ND5 especially in HT-treated ECs (*Figure 4F, G*). To test the hypothesized role of mtRNA in transcriptional activation of the nuclear genome, we performed twofold experiments: (i) smFISH for MT-CYB and co-immunofluorescence (IF) using SC35 antibody which marks nuclear speckles and actively transcribed regions (*Lin et al., 2008*; *Kim et al., 2020*) and (ii) seqFISH for MT-CYB and pre-mRNAs (using intron-targeting probes) transcriptionally induced by HT, that is, VCAM1 and LINC00607 (using probes targeting introns). As shown in *Figure 4H, I*, in HT-treated ECs, a fraction of MT-CYB signals overlapped those of SC35, and the signals of MT-CYB showed proximity to those of HT-induced VCAM1 and LINC00607 pre-mRNAs.

Finally, we sought for evidence that the observed phenomenon and mechanism may occur in vivo and is relevant to health and disease states. We performed smFISH on ECs freshly isolated from the mesenteric arteries of human donors. To correlate with the in vitro experiments, we selected the non-diabetic donors and those with Type 2 diabetes (T2D) according to HbA1C levels and medical record (*Figure 5A*). Of note, the two T2D donors each had 8 and over 20 years of diabetes and were uncompliant to treatments. In agreement with data from cultured ECs, smFISH detected nuclear MT-CYB signals in ECs from all four donors. Furthermore, T2D donor-derived ECs exhibited more nuclear localized MT-CYB puncta compared to non-diabetic donors (*Figure 5B, C*). Together, these data provide supporting evidence for the involvement of mt-caRNAs in endothelial dysfunction in the context of diabetes.

## Discussion

Mitochondrial–nuclear communication is essential for maintaining cellular homeostasis. Mitochondrial–nuclear retrograde signaling can be mediated through an array of molecules including but not limited to metabolites, peptides, and Ca$^+$. These signaling molecules often activate TFs in the cytoplasm, which subsequently translocate into the nucleus to alter transcription. It remains an open question whether any other means of transducing retrograde signals and whether there are other types of molecules that can serve as the messenger.

To be a messenger of retrograde signaling, two criteria must be satisfied. First, this molecule must reflect certain functional status of the mitochondria. Second, this molecule must either directly or indirectly alter the epigenome and thus impact the transcription of nuclear genes. In this work, we report a widespread association of mtRNAs with nuclear chromatin. As the transcriptional output of the mitochondrial genome, mtRNA reflects the transcriptional status of the mitochondria and thus satisfies the first criterion. Furthermore, the presence of high glucose and inflammatory cytokines induced the chromatin-associated mtRNA (mt-caRNA) in human EC, confirming that the level of mt-caRNA responds to cellular stress. Mt-caRNA establishes a direct connection between mtRNA and the epigenome.

Correlated with the stress-induced increase of mt-caRNA are the transcription levels of nuclear genes including adhesion molecules ICAM1 and VCAM1, and increased monocyte–EC attachment. Depletion of a specific mt-caRNA ameliorated stress-induced transcription of nuclear genes including ICAM1 and VCAM1, and abolished the stress-induced increase in monocyte–EC attachment. Reversely, inducing this mt-caRNA in the normal condition reproduced the stress-induced transcriptional changes and the increased monocyte–EC attachment, confirming a for mt-caRNA in transcriptional regulation. These data nominate mtRNA–chromatin interaction as a means of retrograde signaling.

To ensure our discovered mt-caRNA did not originate from NUMTs, we examined the distribution of the mismatch bases between any mtRNA-aligned sequencing reads and NUMT sequences. Not only that all the several million mtRNA-aligned reads exhibited better alignment to mtRNA than any

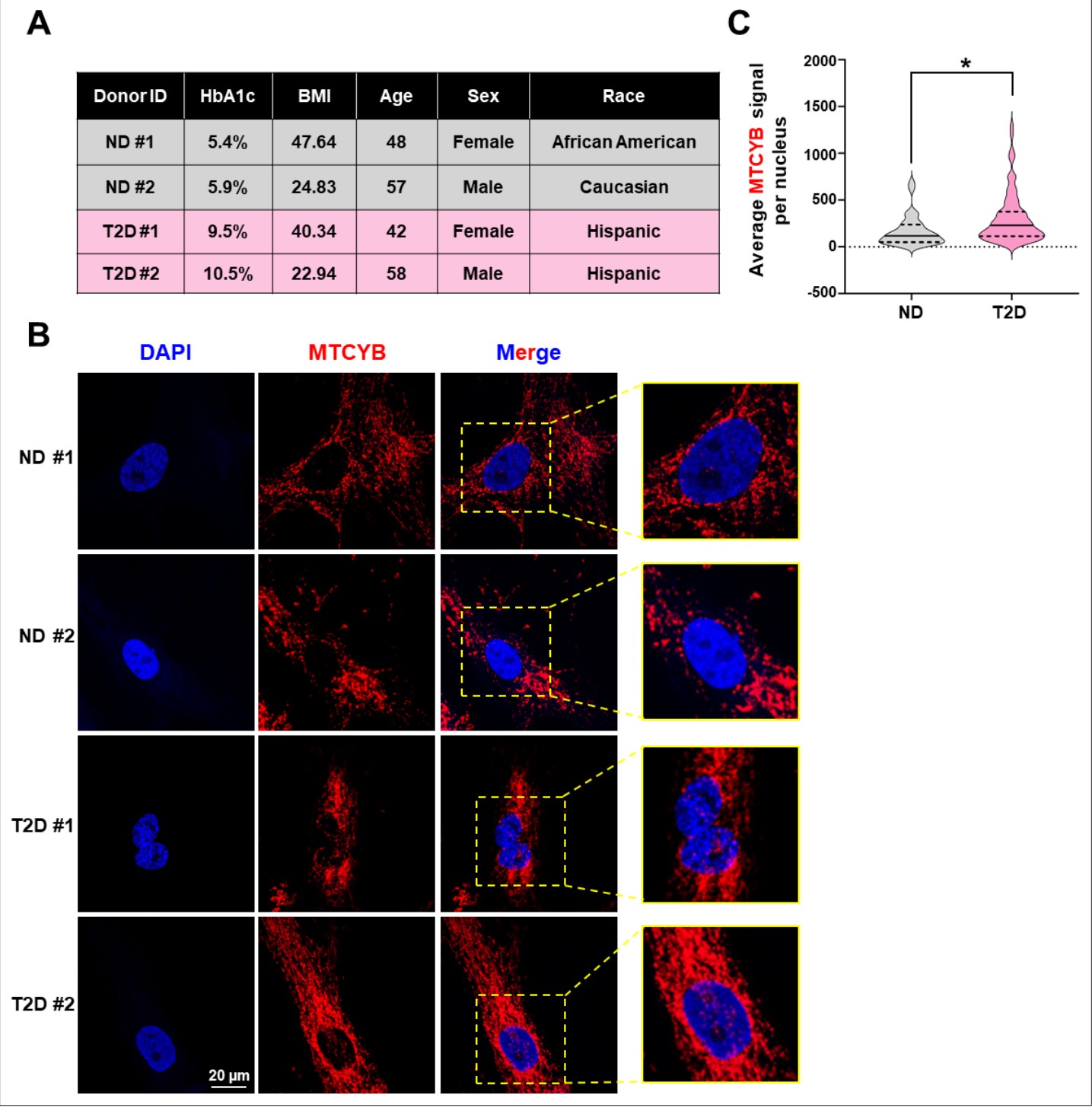

**Figure 5.** Nuclear localization of MT-CYB in donor-derived intimal cells. (**A–C**) Intimal cells were freshly isolated from mesenteric arteries of two non-diabetic (ND) and two Type 2 diabetic (T2D) donors and grown on glass coverslips. (**A**) Table showing donor information. (**B**) smFISH (RNAscope) was performed for MT-CYB transcripts (in red), with nuclear counter stain with DAPI. Representative images from >15 cells per donor are shown. (**C**) Violin plot showing the distribution of average MTCYB signal per nucleus from $n = 34$ for ND and $n = 81$ for T2D. *$p < 0.05$ based on Mann–Whitney $U$-test.

NUMTs, but most of these reads also contain five or more bases of mismatches to any NUMTs. These data suggest that most of mtRNA–nuDNA read pairs are not attributable to possible misalignments. Consistently, to target SncmtRNA, we used BLAST to ensure that the LNA-S GapmeR does not potentially target any homologous transcripts arising from NUMTs. Furthermore, the lack of any detectable

signals for MTCYB and MTND5 in the Rho$^0$ cells also excluded the potential possibility of capturing NUMT-derived transcripts in smFISH. Collectively, these controls should be sufficient to differentiate mtRNA from NUMT-derived RNA. Future studies on the mechanism of nuclear translocation of mtRNA are warranted.

Using SncmtRNA and ECs as an exemplar model to investigate the role of mt-caRNA in nuclear transcription, we observed a strong effect of Snc-KD (primarily in the nucleus) on nuclear transcriptome but not on mitochondrial gene expression and function. At basal (NM) condition, Snc-KD led to a strong suppression of anti-viral and Type 1 interferon response pathways in ECs. While similar effect was observed in HT-stressed EC, Snc-KD also suppressed multiple inflammatory response pathways. These findings support a positive regulation of mt-caRNA in innate immune and inflammatory response and are in line with the well-recognized role of mitochondria in innate immunity (*West et al.,* *2011*). In addition to the release of mtDNA and dsRNA by mitochondria into cytoplasm, mt-caRNA (exemplified by SncmtRNA) may also participate in the mitochondria-controlled intrinsic immune surveillance by transducing the defense signal to the nucleus. This mechanism may also 'prime' the cells for activation of the inflammatory response under pathological stimuli, for example diabetes-mimicking HT or bacterial infection. Indeed, Snc-KD also attenuated LPS-induced inflammatory gene expression, including DDX58, VCAM1, and ICAM1 (*Figure 2—figure supplement 5*). Of note, Snc-KD also had a strong effect in upregulating many cell cycle regulators, as revealed by bulk RNA-seq. However, these genes were not significant DEGs in the snRNA-seq data, suggesting that the effect of SncmtRNA on cell cycle may not be primary. It will be interesting to dissect in future studies the detailed molecular mechanisms underlying mt-caRNA-modulated nuclear transcription.

Finally, using smRNA FISH we showed that MT-CYB and MT-ND5, two other mt-caRNAs are localized in the nucleus of freshly isolated human ECs. Furthermore, their nuclear signals are increased in ECs from diabetic donors and co-localize with active transcription foci, providing in vivo and human disease relevance of mt-caRNAs. A limitation of our study is that these experiments were performed for a few mtRNAs and are with relatively low number of human donor mesenteric arteries. Future studies in ECs from different vascular beds and disease conditions are needed to generalize our findings. As suggested by iMARGI data, the chromatin attachment of mt-caRNAs is likely a more widespread phenomenon in various cell types. Given the critical nuclear–mitochondria crosstalk in most cellular functions and biological processes, our study unravels a previously unidentified mechanism by which mt-caRNA forms an additional layer for nuclear transcriptional and epigenetic regulation, which may have important implications in health and disease.

## Materials and methods
### iMARGI data analyses

iMARGI sequencing reads were processed by the iMARGI-Docker software to identify the RNA–DNA read pairs including nuRNA–nuDNA and mtRNA–nuDNA read pairs (*Wu et al., 2019*). The CAL in terms of normalized CPM was obtained by dividing the number of RNA–nuDNA read pairs involving this type of RNA by the number of RNAs involved in this RNA type and by the number of total RNA–DNA read pairs in the library. The CAL of each type of RNA on any type of genomic region, for example promoters, was obtained by dividing the number of RNA–nuDNA read pairs involving this type of RNA and promoter DNA sequences by the total length of promoters (in kb) and by the number of RNA–nuDNA read pairs involving this type of RNA.

### Cell culture, transfection, and infection

HUVECs tested negative for mycoplasma contamination (Cell Applications, 200-05n) were cultured in complete M199 medium supplemented with 20% fetal bovine serum (FBS; Sigma, M2520) and 1× antibiotics (penicillin–streptomycin, Gibco, 15140122). EC identity was verified by testing for expression of endothelial markers by RT-qPCR and IF. For osmolarity control (NM) samples, 25 mM D-mannitol (Fisher Scientific, M120-500) was added. For HT-treated samples, 25 mM D-glucose + 5 ng/ml TNFα (Thermo Fisher Scientific, PHC3015) were added. HEK-293 and Rho$^0$ cells were cultured with DMEM (Corning 10-013-CV) supplemented with 10% FBS. For imaging experiments, ECs were cultured on coverslips (Bioptechs) pre-coated with 10 mg/ml fibronectin (Sigma F2006) and 0.01% poly-L-lysine (Millipore, A-005-C).

For SncmtRNA knockdown (Snc-KD) experiments, the LNA GapmeRs with scramble sequence or those targeting SncmtRNA (QIAGEN; sequence shown in *Supplementary file 1*) were transfected into HUVECs with Lipofectamine RNAiMAX and Opti-MEM (Thermo Fisher Scientific 13778150) following the manufacturer's guidelines. Four to six hours after transfection, EC media were changed to complete M199 media containing with NM or HT conditions.

The SncmtRNA full-length cDNA sequence was synthesized by Genscript Inc and subcloned into pCMV-MCS packaging vector using ClaI and BglII sites. The control AAV vector, pAAV-CAG-tdTomato (Plasmid #59462) was a gift from Edward Boyden and obtained from Addgene (Addgene plasmid # 59462). HEK293A cells were co-transfected with RepCap, pHelper, and the SncmtRNA/tdTomato plasmids with PEI at 80% confluency. The cells and media were harvested after 96 hr and centrifuged at 1000 rpm for 10 min at 4°C. The pellets were resuspended in phosphate-buffered saline (PBS) containing 0.0001% pluromic F68 and 200 mM NaCl and freeze–thawed in liquid nitrogen. The cell debris were pelleted at 3220 g for 15 min at 4°C and the supernatant containing the cleared lysate and virus was filtered through a 0.45-µm PES membrane before resuspending in PEG solution.

The IRF3/IRF7 luciferase reporter was constructed by subcloning 3× IRF3/7 binding element (GTCA GGAGAAGGAAACCTTC) into the Sal I and HindIII sites in the pGL3-basic (Promega E1751) backbone vector. HEK-293 cells were co-transfected with Renilla and IRF3/7 using Lipofectamine 2000 (Thermo Fisher Scientific) following the manufacturer's protocol. After 48 hr, cells were lysed using Reporter Lysis Buffer (Promega) and the luciferase activity was measured using the Luciferase Assay System (Promega, Cat # E4030) following the manufacturer's guidelines.

## FISH for SncmtRNA

FISH for SncmtRNA was performed using a FAM-labeled LNA-S probe (sequence shown in *Supplementary file 1*) following Stellaris RNA FISH protocol for adherent cells available online at https://www.biosearchtech.com/support/resources/stellaris-rna-fish/stellaris-protocols/ with slight modifications. A FAM-labeled LNA probe with scrambled sequence was used a control. Briefly, cells were grown and fixed on coverslips using 4% paraformaldehyde (PFA). Cells were permeabilized in 70% ethanol for 1 hr followed by hybridization with the probe (200 nM final concentration) in the Stellaris hybridization buffer overnight at 37°C. Subsequently, cells were washed with Stellaris wash buffers A and B, stained with DAPI, and mounted for imaging. Images were taken on Zeiss LSM 880 in Airyscan mode.

## RNA extraction and quantitative PCR

Total RNA was isolated from cells using TRIzol according to the manufacturer's protocol. TURBO DNase (Thermo Fisher Scientific, AM2238) was used to eliminate genomic DNA before reverse transcription with SuperScript IV Reverse Transcriptase (Thermo Fisher Scientific, 18090010) using random hexamer primers. qPCR was performed with Bio-Rad SYBR Green Supermix on Bio-Rad CFX Connect Real Time system. For detecting MT-RNR2-derived transcripts, the extension time per cycle was increased from the standard 30 to 45 s. All primer sequences used for qPCR amplification are listed in *Supplementary file 1*.

## Subcellular fractionation

To isolate RNA from the nuclear and cytoplasmic fractions, HUVECs were collected in 1× PBS using a cell scraper. The cells were pelleted at 300 × *g* for 10 min, resuspended in nuclear isolation buffer (1.28 M sucrose, 40 mM Tris–HCl pH 7.5, 20 mM MgCl$_2$, 4% Triton X-100, 1× protease inhibitor, 1× phenylmethylsulfonyl fluoride (PMSF), and recombinant RNase inhibitor), and incubated on ice for 20 min while vortexing every 5 min. After 20 min, the cell lysates were centrifuged at 2500 × *g* for 15 min in a swing bucket centrifuge. The supernatant containing the cytoplasmic fraction and the pellet containing the nuclear fraction were used to isolate RNA using TRIzol LS and TRIzol, respectively. RNA isolation from nucleoplasmic and chromatin fraction was performed as described previously (*Calandrelli et al., 2020*).

To isolate RNA from the mitochondrial fraction, we followed a protocol described by *Dhir et al., 2018* using MACS mitochondrial isolation kit (Miltenyi Biotec, #130-094-532). Briefly, cell pellets were resuspended in 500 µl lysis buffer provided with the kit, homogenized using a 25 G needle, made up to 10 ml with 1× separation buffer, and incubated with 50 µl TOM-22 beads for 1 hr with rotation at

4°C. Following the incubation, the bound fractions were eluted into 1 ml separation buffer using the MACS LS columns and centrifuged at 14,000 × $g$ for 15 min, followed by RNA extraction using TRIzol.

## Bulk RNA-sequencing and analysis

Total RNA libraries were prepared using the KAPA mRNA HyperPrep Kit (Roche Diagnostics) following the manufacturer's manual. For analysis, raw sequencing reads were processed using Trim Galore and the trimmed reads were aligned to the hg38 reference genome using HISAT2 (*Pertea et al., 2016*). HTseq (*Anders et al., 2015*) was used to generate counts and DESeq2 (*Love et al., 2014*) was then used to perform differential gene expression analysis with default parameters (genes with p-values <0.05 and Log$_2$ fold-change > |0.5| were considered significant differently expressed genes).

## Monocyte adhesion assay

Leukapheresis products and discards were obtained from consented research participants (healthy donors) under protocols approved by the City of Hope Internal Review Board (IRB #09025). Peripheral blood mononuclear cells (PBMCs) were isolated by density gradient centrifugation over Ficoll-Paque (GE Healthcare) followed by multiple washes in PBS/ethylenediaminetetraacetic acid (EDTA) (Miltenyi Biotec). Cells were rested overnight at room temperature on a rotator, and subsequently washed and resuspended in complete X-VIVO. Up to 5 × 10$^9$ PBMC were incubated with anti-CD14 microbeads (Miltenyi Biotec) for 30 min at room temperature and magnetically enriched using the CliniMACS system (Miltenyi Biotec) according to the manufacturer's protocol. The monocytes were labeled with CellTracker Green CMFDA Dye (Thermo Fisher Scientific) and incubated with monolayer HUVECs (4 × 10$^3$ cells per cm$^2$) for 15–30 min in a cell culture incubator. The nonattached monocytes were then washed off with complete EC growth medium. The attached monocyte numbers were quantified on Cytation1 Cell Imaging Multi-Mode Reader (BioTek) using green fluorescent channel. Average numbers per condition were calculated from two randomly selected fields of technical duplicates.

## FUCCI analysis

Cell cycle analysis was performed using the Premo FUCCI Cell Cycle Sensor kit (Thermo Fisher Scientific; Cat # P36237) following the manufacturer's protocol. Briefly, Premo geminin-GFP and Premo Cdt1-RFP were added to the EC culture medium at 80 particles per cell. Cdt1 levels are highest during G1, while Geminin levels are highest during the S, G2, and M phases. Thus, tracking the fluorescence will enable monitoring cell cycle progression overtime. Twenty-four hours later, cells were imaged using Cytation1 Cell Imaging Multi-Mode Reader (BioTek). Green fluorescent channel was used for GFP, and red fluorescent channel for RFP. The number of cells with red fluorescence (indicative of G1 phase), yellow fluorescence (indicative of G1/S transition), and green fluorescence (indicative of G2/M) was counted from at least 4 fields per condition and five independent experiments.

## scRNA- and snRNA-sequencing and analysis

HUVECs treated with HT and transfected with LNA in biological replicates were processed following the Drop-seq protocol from 10× Genomics. For scRNA-seq, ECs were washed, trypsinized, and suspended into a single-cell solution for the library preparation using 10× Genomics Chromium 3.1′ expression kit as previously described (*Calandrelli et al., 2020*). For snRNA-seq, cells were first washed with 1× PBS + 0.04% bovine serum albumin (BSA), pelleted at 300 × $g$ for 5 min, and resuspended in 100 µl of freshly prepared lysis buffer (10 mM Tris–HCl pH 7.4, 10 mM NaCl, 3 mM MgCl$_2$, 0.1% Tween-20, 0.1% Nonidet P40 Substitute, 0.01% Digitonin, 1% BSA). The cells were then lysed on ice for 5 min, checked for nuclear integrity under microscope, and washed three times in wash buffer (10 mM Tris–HCl pH 7.4, 10 mM NaCl, 3 mM MgCl$_2$, 0.1% Tween-20, 1% BSA). The resulting nuclei underwent the Drop-seq protocol with 10× Genomics Chromium 3.1′ expression kit.

scRNA- and snRNA-seq data were processed using the standardized pipeline provided by 10× Genomics. The reads were aligned to both intronic and exonic regions of the human hg38 reference genome by utilizing include-introns command during alignment with Cell Ranger. Seurat R package (v3) was used following well-established filtering steps to remove genes expressed in <3 cells and cells expressing <200 genes. For the differential analysis between HT-Scr and HT-LNA conditions, rare cells/nuclei with very high numbers of genes (>8500 and 7000 genes for scRNA- and snRNA-seq, respectively; potential multiplets) and cells/nuclei with high mitochondrial read percentages (>15% for

scRNA-seq and >25% for snRNA-seq) were removed. HT_Scr and HT_LNA samples were first separately normalized with 'sctransform' and then integrated into one dataset based on Seurat-selected anchor genes. This integrated dataset was used for clustering analysis. Clusters of nuclei were identified by Seurat based on shared nearest neighbor using the first 10 principal components as input resolution = 0.5. Differential expression analysis was performed using the Wilcoxon test with default parameters in Seurat. The threshold for the avg_logFC was set to be 0.1.

## Nascent RNA pull-down

Nascent RNA pull-down assay was performed as previously reported (*Calandrelli et al., 2020*). Briefly, ECs were synchronized by serum starvation with M199 + 2% FBS for 6 hr. Nascent/newly synthesized RNA were labeled with 5-ethynyl uridine (EU) at 200 μM final concentration and incubated for 24 hr. Total RNA was then isolated from the cells and 3 μg of total RNA was used to capture nascent RNA using the Click-iT Nascent RNA Capture Kit (Thermo Fisher Scientific, C10365) following the manufacturer's protocol.

## smFISH, co-smFISH-IF, and confocal and super-resolution microscopy

smFISH shown in *Figure 4A–E, H* and *Figure 4—figure supplement 1* were performed using RNAscope Multiplex Fluorescent V2 Assay (ACDBio, 323100) following the manufacturer's protocol. Briefly, cells were fixed with 4% PFA for 30 min at room temperature, ethanol-dehydrated, pre-treated with hydrogen peroxide ($H_2O_2$) for 10 min at room temperature, and permeabilized with Protease III (1:10 dilution) for 30 min at room temperature prior to probe hybridization following recommended protocol. RNAscope Probe – Hs-MT-CYB (Cat # 582771) and RNAscope Probe – Hs-MT-ND5-C2 (Cat # 539451-C2) were used to detect human mitochondrial MT-CYB and MT-ND5 transcripts, respectively. Following hybridization, the RNAscope assay was developed following the manufacturer's protocol. Opal dyes 570 or 690 were used to detect the mitochondrial transcripts. To image mitochondria and mitochondrial transcripts simultaneously, cells were incubated with MitoTracker Deep Red FM (Thermo Fisher Scientific, M22426) at 100 nM final concentration in serum-free media for 30 min prior to fixing.

smFISH combined with IF was performed using the RNA-Protein Co-Detection Ancillary Kit (ACDBio, Cat # 323180) along with the RNAscope Multiplex Fluorescent V2 Assay (ACDBio, Cat # 323100) following the manufacturer's protocol. Briefly, cells were fixed as in smFISH and blocked with co-detection antibody diluent for 30 min at room temperature prior to incubating with anti-human SC35 (Abcam, Cat # ab11826) at 1:50 dilution overnight at 4°C. Cells were washed twice with 1× PBST (1× PBS $Ca^{2+}$ and $Mg^{2+}$ free + 0.1% Tween-20) and fixed again with 4% PFA before proceeding with permeabilization and RNAscope assay as abovementioned. After completing the final step of the RNAscope assay, the cells were incubated with secondary antibody anti-mouse Alexa 488 (Thermo Fisher Scientific, Cat # A-11008) at 1:200 dilution for 30 min at room temperature, washed twice with 1× PBST and mounted for imaging. DAPI was used to stain the DNA in the nuclei.

Confocal microscopy was performed on Zeiss LSM 700 microscope using ×20 or ×63 water immersion lens. 3D reconstruction of confocal images was performed using Imaris software with 'Surfaces Technology'. Super-resolution images were captured on Zeiss LSM 880 microscope using ×63 oil immersion lens in the Airyscan mode.

Image analysis for smFISH was performed using FIJI (ImageJ software) in a blinded fashion. A mask was drawn around the nuclear region (as defined by DAPI channel), and the average gray value of MTCYB signals within this mask was used to compute MTCYB signals per nucleus. Image analysis for mitochondrial network quantification was performed using ImageJ using a network analysis tool (*Chaudhry et al., 2020*).

smFISH shown in *Figure 4F, G, I* were performed following a previously published protocol (*Liu et al., 2020*) with modifications. The probes were designed using ProbeDealer (*Hu et al., 2020*) and synthesized by IDT. ProbeDealer design is based on melting temperature, GC content, exclusion of repetitive sequences, and the specificity was further confirmed by BLAST. Each probe consists of a 30-nt target region that specifically binds to transcript of interest and a 20-nt readout region that allows the binding of dye-labeled secondary oligos as previously validated (*Liu et al., 2020*; *Chen et al., 2015*). Specifically, 48 oligos for VCAM and LINC00607 and 32 oligos for MT-CYB and 45 oligos for MT-ND5 were used.

Cells were fixed with 4% PFA (EMS, 15710), permeabilized with 0.5% Triton X-100 (Sigma, T8787), and incubated in the pre-hybridization buffer with 50% formamide (Sigma, F7503) in 2× saline-sodium citrate (SSC, Invitrogen 15557-044) for 5 min at RT and then in the hybridization buffer containing 50% formamide, 0.1% yeast tRNA (Life Technologies, 15401011), 10% dextran sulfate (Millipore, S4030), and 1% murine RNase inhibitor (New England Biolabs, M0314L) in 2× SSC with 6 nM primary probes at 37°C for 24 hr. Samples were then washed with 2× SSC containing 0.1% Tween 20 at 60°C, followed by 2× SSCT wash at RT. The sample then underwent multiple rounds of sequential imaging, using Translura's automated sequential smFISH microscope, and corresponding secondary hybridization, washing, and imaging buffers. Final concentration of the secondary oligos were 3.75 nM. Z-stack images were taken with 750 or 647 nm laser, with a step size of 600 nm and total Z-stacks range of 10 μm.

## Seahorse assay

HUVECs were assayed using the Seahorse XFe24 Analyzer (Agilent Technologies, Lexington, MA) according to the manufacturer's instructions. Briefly, the ECs transfected with either scrambled or LNA-S targeting SncmtRNA were plated at 20,000 cells/well 48 hr post transfection and allowed to adhere overnight. The day of the assay, media was changed to Seahorse XF DMEM base medium without phenol red (Agilent, 103334-100) supplemented with 10 mM glucose, 2 mM glutamine, and 1 mM sodium pyruvate pH 7.4. OCR was measured using the XF Cell Mito Stress Test Kit (Agilent, 103015-100) following injections of 1.5 μM oligomycin, 1 μM FCCP, and 0.5 μm rotenone and antimycin.

## Donor-derived ECs

Human mesenteric artery tissue studies were conducted on deidentified specimens obtained from the Southern California Islet Cell Resource Center at City of Hope. The research consents for the use of postmortem human tissues were obtained from the donors' next of kin and ethical approval for this study was granted by the Institutional Review Board of City of Hope (IRB #01046). T2D was identified based on diagnosis in the donors' medical records and the percentage of glycated hemoglobin A1c (HbA1c) of 6.5% or higher. Donor-derived ECs were isolated following our published protocol (*Malhi et al., 2022*). Briefly, the mesenteric arteries were cut open lengthwise to expose the lumen. The intimal layer containing ECs was scraped mechanically using a scalpel in pre-warmed digestion buffer containing Collagenase D. The collected intimal cells were digested for 5 min at 37°C, washed with complete M199 medium, and then plated onto tissue culture dishes coated with attachment factor or collagen. Upon cell attachment and growth, the cells were transferred onto a cover glass for smFISH experiments.

## Statistical analysis

For all experiments, at least three independent experiments were performed. Statistical analysis was performed as indicated in the results and methods, or by using Student's *t*-tests for two group comparisons or ANOVA with post hoc tests as appropriate for multiple group comparisons, unless otherwise indicated. For all the high-throughput sequencing data, experiments were performed in two to three biological replicates. $p < 0.05$ was considered statistically significant unless otherwise indicated.

## Acknowledgements

The authors thank Dr. John Burnett at City of Hope for providing Rho$^0$ cells, Drs. Ismail Al-Abdullah and Meirigeng Qi of the islet transplantation team at City of Hope for isolation of human arteries, Drs. John Y-J. Shyy, John Rossi, Yilun Liu, and John Burnett for helpful discussion and valuable inputs, and Dr. Siyuan Wang at Yale University for technical consultation on smFISH.

## Additional information

### Competing interests

Ji Shi: JS is a co-founder of Translura, Inc. Sheng Zhong: SZ is a founder of Genemo, Inc. The other authors declare that no competing interests exist.

## Funding

| Funder | Grant reference number | Author |
| --- | --- | --- |
| National Institutes of Health | R01HL145170 | Zhen Bouman Chen |
| National Institutes of Health | R01GM138852 | Sheng Zhong |
| National Institutes of Health | R01HD107206 | Sheng Zhong |
| National Institutes of Health | DP1DK126138 | Sheng Zhong |
| National Institutes of Health | R01HL106089 | Rama Natarajan |
| Ella Fitzgerald Foundation | | Zhen Bouman Chen |
| California Institute of Regenerative Medicine | EDU4-12772 | Alonso Tapia |
| National Institutes of Health | P30CA033572 | Brian Armstrong |
| Kruger research grant | | Sheng Zhong |

The funders had no role in study design, data collection, and interpretation, or the decision to submit the work for publication.

## Author contributions

Kiran Sriram, Conceptualization, Data curation, Formal analysis, Validation, Investigation, Visualization, Methodology, Writing – original draft, Project administration, Writing – review and editing; Zhijie Qi, Data curation, Software, Formal analysis, Validation, Investigation, Visualization, Methodology, Writing – original draft, Writing – review and editing; Dongqiang Yuan, Formal analysis, Investigation, Writing – review and editing; Naseeb Kaur Malhi, Formal analysis, Investigation, Methodology, Writing – review and editing; Xuejing Liu, Yingjun Luo, Investigation, Methodology, Writing – review and editing; Riccardo Calandrelli, Software, Formal analysis; Alonso Tapia, Investigation, Methodology; Shengyan Jin, Ji Shi, Martha Salas, Methodology; Runrui Dang, Jiayu Liao, Resources; Brian Armstrong, Supervision; Saul J Priceman, Ping H Wang, Writing – review and editing; Rama Natarajan, Supervision, Funding acquisition, Writing – review and editing; Sheng Zhong, Zhen Bouman Chen, Conceptualization, Resources, Data curation, Supervision, Funding acquisition, Visualization, Writing – original draft, Project administration, Writing – review and editing

## Author ORCIDs

Kiran Sriram ⓘ https://orcid.org/0000-0002-7412-2380
Zhijie Qi ⓘ http://orcid.org/0000-0002-0022-8844
Naseeb Kaur Malhi ⓘ http://orcid.org/0000-0002-8981-2974
Xuejing Liu ⓘ http://orcid.org/0000-0002-3931-0191
Riccardo Calandrelli ⓘ http://orcid.org/0000-0002-4541-6320
Shengyan Jin ⓘ http://orcid.org/0000-0002-8667-7586
Martha Salas ⓘ http://orcid.org/0009-0006-0159-5896
Brian Armstrong ⓘ http://orcid.org/0009-0003-5026-2631
Saul J Priceman ⓘ http://orcid.org/0000-0002-8136-2112
Ping H Wang ⓘ http://orcid.org/0000-0002-7847-7752
Rama Natarajan ⓘ http://orcid.org/0000-0003-4494-1788
Zhen Bouman Chen ⓘ https://orcid.org/0000-0002-3291-1090

## Ethics

Human mesenteric artery tissue studies were conducted on deidentified specimens obtained from the Southern California Islet Cell Resource Center at City of Hope. The research consents for the use of postmortem human tissues were obtained from the donors' next of kin and ethical approval for this study was granted by the Institutional Review Board of City of Hope (IRB #01046).

**Decision letter and Author response**
Decision letter https://doi.org/10.7554/eLife.86204.sa1
Author response https://doi.org/10.7554/eLife.86204.sa2

## Additional files

### Supplementary files
• Supplementary file 1. RT-qPCR and locked nucleic acid (LNA) GapmeR sequences.
• Supplementary file 2. Differentially expressed genes from bulk and single-cell RNA-seq upon SncmtRNA knockdown and overexpression.
• MDAR checklist

### Data availability
All data needed to evaluate the conclusions in the paper are present in the paper and/or supporting files. All high-throughput sequencing data used in this study have been deposited at GEO with accession number GSE211971.

The following dataset was generated:

| Author(s) | Year | Dataset title | Dataset URL | Database and Identifier |
|---|---|---|---|---|
| Sriram K, Qi Z, Yuan D, Malhi NK, Liu X, Calandrelli R, Luo Y, Jin S, Shi J, Salas M, Dang R, Armstrong B, Priceman SJ, Wang P, Liao J, Natarajan R, Zhong S, Chen ZB | 2024 | Regulation of Nuclear Transcription by Mitochondrial RNA | https://www.ncbi.nlm.nih.gov/geo/query/acc.cgi?acc=GSE211971 | NCBI Gene Expression Omnibus, GSE211971 |

The following previously published datasets were used:

| Author(s) | Year | Dataset title | Dataset URL | Database and Identifier |
|---|---|---|---|---|
| Calandrelli R, Xu L, Luo Y, Wu W, Fan X, Nguyen T, Chen CJ, Sriram K, Tang X, Burns AB, Natarajan R, Chen ZB, Zhong S | 2020 | Stress-induced RNA-chromatin interactions promote endothelial dysfunction | https://www.ncbi.nlm.nih.gov/geo/query/acc.cgi?acc=GSE135357 | NCBI Gene Expression Omnibus, GSE135357 |
| Calandrelli R, Wen X, Nguyen TC, Chen CJ, Qi Z, Chen W, Yan Z, Wu W, Zaleta-Rivera K, Hu R, Yu M, Wang Y, Ma J, Ren B, Zhong S | 2021 | Three-dimensional organization of chromatin associated RNAs and their role in chromatin architecture in human cells | https://data.4dnucleome.org/experiment-set-replicates/4DNESNOJ7HY7/ | 4D Nucleome Data Portal, 4DNESNOJ7HY7 |
| Calandrelli R, Wen X, Nguyen TC, Chen CJ, Qi Z, Chen W, Yan Z, Wu W, Zaleta-Rivera K, Hu R, Yu M, Wang Y, Ma J, Ren B, Zhong S | 2021 | Three-dimensional organization of chromatin associated RNAs and their role in chromatin architecture in human cells | https://data.4dnucleome.org/experiment-set-replicates/4DNES9Y1GHK4/ | 4D Nucleome Data Portal, 4DNES9Y1GHK4 |
| Calandrelli R, Wen X, Nguyen TC, Chen CJ, Qi Z, Chen W, Yan Z, Wu W, Zaleta-Rivera K, Hu R, Yu M, Wang Y, Ma J, Ren B, Zhong S | 2021 | Three-dimensional organization of chromatin associated RNAs and their role in chromatin architecture in human cells | https://data.4dnucleome.org/experiment-set-replicates/4DNESIKCVASO/ | 4D Nucleome Data Portal, 4DNESIKCVASO |

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
