## [Editor Report]

This work is fundamental in providing compelling evidence of mitochondria-encoded RNAs playing a role in controlling nuclear gene expression. How mitochondria and the nucleus communicate is an important yet, not well-appreciated area of biology. Using the iMARGI (in situ mapping of RNA-Genome Interactions) technology developed by this team, the authors found that mitochondria-encoded RNAs play an unexpected role in regulating nuclear gene expressions in endothelial cells and intriguingly, depletion or overexpression of a specific mt-caRNA altered stress-induced transcription of nuclear genes encoding for innate inflammation and endothelial activation. Overall, these findings are interesting and supported by experimental confirmation, bulk-RNA-seq, and snRNA and scRNA-seq data and will be of interest to the field studying RNA regulation, gene expression, and cell biology.

---

## [Decision Letter]

**Decision letter after peer review:**

Thank you for submitting your article "Regulation of Nuclear Transcription by Mitochondrial RNA" for consideration by *eLife*. Your article has been reviewed by 2 peer reviewers, and the evaluation has been overseen by a Reviewing Editor. The reviewers have opted to remain anonymous.

Essential revisions:

1. Can the authors clarify if mt-caRNAs is enriched at the promoters of genes that respond to HT stress conditions?

2. Please clarify if Snc-KD affects mitochondrial function at the phenotypic level? It would also be useful to measure changes in mt-caRNAs expression due to HT stress treatment.

*Reviewer #1 (Recommendations for the authors):*

The manuscript titled "Regulation of Nuclear Transcription by Mitochondrial RNA" by Sriram et al. explores the three-dimensional organization of chromatin-associated RNAs and their role in regulating chromatin architecture in multiple human cell types. The authors developed iMARI (in situ mapping of RNA-Genome Interactions) technology and found that mitochondria-encoded lncRNA plays a role in regulating nuclear gene expressions. They then performed experimental confirmation, bulk-RNA-seq, snRNA, and scRNA-seq to demonstrate the function of SncmtRNA in regulating nuclear gene expression in EC cells. The manuscript is written and organized very well.

*Reviewer #2 (Recommendations for the authors):*

This is an interesting study on the role of mt-caRNAs in mitochondrial-nuclear crosstalk.

1. The preferential genomic association with promoters has been shown in multiple cell types. The authors have also done an elegant experiment showing stress-induced transcription of inflammatory genes in endothelial cells. Is it through enriched association of SncmtRNA with promoters in response of HT stress condition?

2. While it remains to be determined how mt-caRNAs translocate into nucleus, it is quite surprising to observe low fraction % of chromatin-associated SncmtRNA in the nucleoplasm and cytoplasm at steady state. Any interpretation of this?

3. The findings from Snc-KD experiments that affect genes involved in innate immune and inflammatory response are very intriguing. Generally, SncmtRNA has been proposed to contribute to the maintenance of mtDNA integrity. Dysregulation of mitochondrial RNA can lead to impaired ATP production, which can affect endothelial cell metabolism and function. Does Snc-KD affect mitochondrial function at the phenotypic level?

4. Changes in SncmtRNA expression may reflect a response to maintain mitochondrial homeostasis and cellular function. Have they measured changes in SncmtRNA expression due to HT condition?

5. We recognize the difficulty in obtaining human vascular biopsy. There are heterogeneities in the response of vascular beds in different organ to diabetic/ inflammatory conditions. Mesenteric arteries, used here, have been observed to be less prone to atherosclerosis compared to large central arteries like the coronary or carotid arteries. Hence, the increased mt-caRNA nuclear signals in endothelial cells from diabetic donors, and their co-localization with active transcription loci may not have proven that it is a common phenomenon present in all cell types, nor is it a validation of pathogenic response.

---

## [Author Response]

Essential revisions:1. Can the authors clarify if mt-caRNAs is enriched at the promoters of genes that respond to HT stress conditions?

Comparing the RNA-seq data in HT and baseline (NM) conditions, we identified 312 genes with higher expression in HT (HT-induced genes) (adjusted p-value<0.05). These HT-induced genes included pro-inflammatory and extracellular matrix remodeling genes.

We compared the chromatin-association levels (CAL) of all mtRNAs in the promoters of HT-induced genes against the other promoters, and other genomic regions including enhancers, super enhancers, introns, 5UTRs, 3UTRs, coding sequences (CDS), and intergenic regions. The promoters of HT-induced genes exhibited higher CAL of mtRNAs than all other types of genomic regions under NM condition (p=0.029, student’s t test). In line with the overall mtRNA CAL, that of SncmtRNA is also higher in the promoters of HT-induced genes than the other genomic regions under normal condition (p=0.016, student’s t test). We observed the same behaviors for both mtRNA and SncmtRNA under HT condition with slightly higher statistical significance (p=0.025 for mtRNA, p=0.0087 for SncmtRNA, student’s t test). These results indicate that mt-caRNAs including SncmtRNA are indeed enriched at the promoters of genes that respond to HT conditions under both normal and HT conditions. We have included these data in the revised Figure 1L-O.

2. Please clarify if Snc-KD affects mitochondrial function at the phenotypic level? It would also be useful to measure changes in mt-caRNAs expression due to HT stress treatment.

To address the first question, we have performed following experiments and analyses:

1) Mitochondrial morphology (mitotracker staining): we have performed mitotracker staining as shown in the original Figure Supplement 2. We have now quantified the number of mitochondrial branches, mitochondrial branch length and junction. We have not identified any significant difference between the EC with vs without SncmtRNA-KD.

2) Mitochondrial function – oxygen consumption rate (OCR) (Seahorse assay): We did not observe a consistent effect of Snc-KD in OCR.

3) SncmtRNA-KD did not have any significant effect on the levels of mtRNAs, or those of nuclear-encoded genes known to regulate mitochondrial biogenesis, either under NM or HT.

Collectively, Snc-KD does not seem to affect mitochondrial function at the phenotypic level. We have included all these data to Figure 2, figure supplement 3 in the revised manuscript.

We have quantified the mtRNAs that show chromatin association (i.e., mt-caRNA) under HT based on RNA-seq. There is no clear effect by HT on the levels of these mt-caRNAs. This has been added to Figure 2 —figure supplement 1F.

Reviewer #2 (Recommendations for the authors):This is an interesting study on the role of mt-caRNAs in mitochondrial-nuclear crosstalk.1. The preferential genomic association with promoters has been shown in multiple cell types. The authors have also done an elegant experiment showing stress-induced transcription of inflammatory genes in endothelial cells. Is it through enriched association of SncmtRNA with promoters in response of HT stress condition?

From the iMARGI and RNA-seq data from ECs under NM and HT, we identified 14 HT-induced genes that exhibit significant increase in the promoter-associated SncmtRNA (Benjamini-Hochberg adjusted p-value<0.05). To evaluate whether these genes are more likely to be downregulated by SncmtRNA knockdown (Snc-KD), we performed an association (Chi-square) test between these genes and the Snc-KD-reduced genes. Out of these 14 genes, Snc-KD reduced the expression levels of 12 of them under HT (Snc-KD vs Scr fold-change < 1), including 6 (*ICAM1, ZFHX2, ABCA6, CSF1, PDE5A* and *RND1)* with statistical significance (Benjamini-Hochberg adjusted p-value<0.05), suggesting the genes suppressed by Snc-KD are enriched in these genes (odds ratio=2.82, p-value=0.0096, Chi-square test). These data reveal an enriched association between genes that show increased SncmtRNA-promoter association and genes that are positively regulated by SncmtRNA (i.e., Snc-KD-reduced genes) under HT. This supports the thesis that HT-induced transcription of a set of pro-inflammatory (and extracellular matrix remodeling) genes is likely through enriched association of SncmtRNA with their promoters. We have added these new data to Figure 2—figure supplement 4.

2. While it remains to be determined how mt-caRNAs translocate into nucleus, it is quite surprising to observe low fraction % of chromatin-associated SncmtRNA in the nucleoplasm and cytoplasm at steady state. Any interpretation of this?

While the fraction % of SncmtRNA in the nucleoplasm at steady state is about 50%, that in the cytoplasm is indeed surprisingly low. Our interpretation, largely speculative, is the following:

1) Unlike in the nucleus, where SncmtRNA can bind to chromatins, SncmtRNA may not have binding partners (e.g., proteins or DNA) in the cytoplasm, rendering the transcript more prone to degradation; 2) SncmtRNA, without any coding potential to support the mitochondrial structure/function per se, may mainly serve as a messenger from mitochondria to communicate with the nucleus, where it exerts its regulatory function. As revealed by Snc-KD experiments, the nuclear inhibition of SncmtRNA under NM led to a significant effect in pathways such as Type I inflammatory response and cell proliferation.

3. The findings from Snc-KD experiments that affect genes involved in innate immune and inflammatory response are very intriguing. Generally, SncmtRNA has been proposed to contribute to the maintenance of mtDNA integrity. Dysregulation of mitochondrial RNA can lead to impaired ATP production, which can affect endothelial cell metabolism and function. Does Snc-KD affect mitochondrial function at the phenotypic level?

We did not observe any significant effect by Snc-KD on mitochondrial network as well as phenotypic function, as quantified by mitotracker staining and network analysis (A-F) and quantification of oxygen consumption rate by seahorse assay (G). We have included this data in Figure 2 —figure supplement 3.

4. Changes in SncmtRNA expression may reflect a response to maintain mitochondrial homeostasis and cellular function. Have they measured changes in SncmtRNA expression due to HT condition?

We have measured the changes in SncmtRNA expression by RT-qPCR under HT conditions and observed no significant differences compared to NM conditions from 5 independent experiments, as shown in Figure 2E.

5. We recognize the difficulty in obtaining human vascular biopsy. There are heterogeneities in the response of vascular beds in different organ to diabetic/ inflammatory conditions. Mesenteric arteries, used here, have been observed to be less prone to atherosclerosis compared to large central arteries like the coronary or carotid arteries. Hence, the increased mt-caRNA nuclear signals in endothelial cells from diabetic donors, and their co-localization with active transcription loci may not have proven that it is a common phenomenon present in all cell types, nor is it a validation of pathogenic response.

We fully agree with the reviewer’s critique and have added this as a limitation of our study in the Discussion.